

# Scattering and Strebel graphs

**Pronobesh Maity**

International Centre for Theoretical Sciences,
Shivakote, Hesaraghatta Hobli, Bengaluru North 560 089, India

pronobesh.maity@icts.res.in

## Abstract

We consider a special scattering experiment with n particles in $\mathbb{R}^{1,n-3}$. The scattering equations in this set-up become the saddle-point equations of a Penner-like matrix model, where in the large $n$ limit, the spectral curve is directly related to the unique Strebel differential on a Riemann sphere with three punctures. The solutions to the scattering equations localize along different kinds of graphs, tuned by a kinematic variable. We conclude with a few comments on a connection between these graphs and scattering in the Gross-Mende limit.

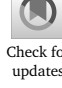
# 1   Introduction

Scattering equations play a key role in the fascinating works of Cachazo, He and Yuan [2–4], which translate the tree-level scattering amplitudes of massless particles, written in terms of combinatoric Feynman diagrams, as a sum over rational functions evaluated at the solutions of those equations and hence reduce it to an algebraic problem. Historically these equations first appeared in the works of Fairlie and Roberts [5–7], in addressing the questions of modifying the Veneziano amplitude to make them tachyon free, and then in the study of high-energy behaviour of string theory by Gross and Mende [17].

It is therefore crucially important to analyse the solutions of this system of equations, and indeed efforts have been made [9–12] to find numerical algorithms to solve the scattering equations for general number of particles, starting from the explicit solutions for few particles. But a good analytical method is still missing. Some other developments include [15, 16].

In this work, we will explicate some new interesting patterns in these solutions for a specific set of kinematics, with a possible connection to the string geometry in the Gross-Mende limit. Before giving a summary of our analysis, we mention some recent developments in a not-so-related field of symmetric product orbifold CFTs, which, interestingly, constitute the main technical support for our work:

Algebraic relations determining the covering maps of the symmetric product CFTs were observed to resemble the scattering equations in [24]. Recently in [1], these covering maps of the CFTs were shown, using Penner-like matrix models, to define the Strebel metric on the "covering space". Interpreting the latter as the dual worldsheet, this gave an explicit demonstration of a mechanism for the gauge-string duality. We will adapt the technology developed there, to apply to a scattering set-up, demonstrating further that there are transitions in the graphs, where the complex solutions of the scattering equations localize, in suitable limits of the kinematics.

In this note, we consider a special scattering experiment with $(N+3)$ massless particles in $\mathbb{R}^{1,d-1}$ where two incoming particles are highly energetic with their energy $E$ scaling with $N$ as $E = \frac{1}{4}N\epsilon$. The $N$ other outgoing particles have the same energy $k^0$ and a common Mandelstam invariant $s$. The special kinematics for these outgoing particles forces the corresponding $N$ momenta to have been directed towards the vertices of a $(N-1)$-simplex from its center in $\mathbb{R}^{N-1}$. This fixes the space-time dimension to be $d = N+1$. Importantly, the restriction to this scattering process will allow us to translate the scattering equation in this set-up to the saddle point equation of a Penner-like matrix model of matrix rank $N$ with three "charges", determined by the kinematic variable $q = \epsilon k^0/s$, located at $(-i)$, $(+i)$ and $\infty$.

We then exploit the standard matrix model technology to find the spectral geometry for this problem; the spectral curve defines a quadratic differential $\phi(z)dz^2$ on the sphere

$$y^2(z)dz^2 = -4\pi^2\phi(z)dz^2,\tag{1.1}$$

and rather astonishingly, as was first pointed out in [1], it essentially becomes the Strebel differential on the sphere with three punctures, for $q < 1$ in our set up. Depending on the range of $q$, the characterizing graphs of this quadratic form take different shapes as in Fig. 1.1.

Having seen how the solutions of the scattering equation localize along these graphs, we then turn the discussion of Gross-Mende strings where we have the same scattering equation determining the high energy, fixed angle behaviour of string amplitudes. For critical space-time dimensions with $N = 25$ or $9$ in our set-up, time component of the space-time trajectories of the Gross-Mende strings become linked with the quadratic differential $\phi(z)dz^2$, in the leading

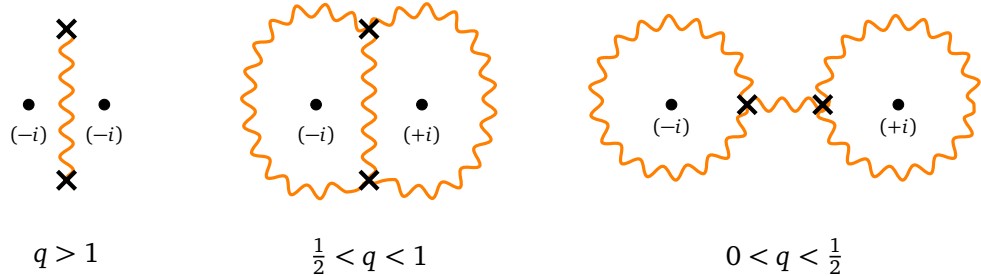

$$q > 1 \qquad \tfrac{1}{2} < q < 1 \qquad 0 < q < \tfrac{1}{2}$$

Figure 1.1: Depending on the kinematic parameter $q = \epsilon k^0/s$, the solutions of the scattering equations localize along different kinds of graphs.

order in $N$, as

$$X^0(z) = \frac{\pi N s}{2 k^0} \int^z \sqrt{\phi(z')}\, dz', \tag{1.2}$$

where $\{X^\mu(z,\bar{z})\}$ describes the trajectory. In fact the string scattering amplitude can also be expressed as an integral along the graphs. These connections might convey the analogues of the transitions of graphs in Fig 1.1 for the string geometry as we tune the kinematic parameter $q$.

Throughout the paper, we will assume $(-+\cdots+)$ convention.

## 2 Scattering set-up

In this section, we outline the basic set-up of the scattering process and corresponding scattering equations. A somewhat similar construction was also discussed in [14] with real solutions for the scattering equations, whereas we will mainly concentrate on the regime of complex solutions with further specialization in the kinematic configuration of the particles.

The explicit scattering experiment in $\mathbb{R}^{1,d-1}$ involves two incoming particles $A$ and B with their momenta as,

$$k_A = (E, 0, \cdots, 0, -E), \quad k_B = (E, 0, \cdots, 0, E), \qquad E > 0, \tag{2.3}$$

where $E > 0$ ensures that the particles are incoming, and $N$ outgoing scattered particles indexed by $a$ with

$$k_a = (k_a^0, k_a^1, \cdots, k_a^{d-2}, 0) \quad a = 1, 2, \cdots, N, \tag{2.4}$$

with $k_a^0 < 0$, since outgoing. From momentum conservation, the other particle $C$ has the momentum

$$k_C = \left(-2E - \sum_{a=1}^N k_a^0, -\sum_{a=1}^N k_a^1, \cdots, -\sum_{a=1}^N k_a^{d-2}, 0\right). \tag{2.5}$$

This particle C will be incoming/outgoing depending on the sign of $(-2E - \sum_{a=1}^N k_a^0)$. The on-shell condition on $k_a$ implies $k_a^0 = -|\vec{k}_a|\ \forall a$, and similar constraint applies to $k_C$. It's clear that $(N+1)$ particles $\{a\}$ and $C$ are confined in the hyperplane $\mathbb{R}^{d-2}$, perpendicular to the incoming particles $A$ and $B$, as in Fig 2.2. Also the momenta of these $(N+1)$ particles yield $s_{aA} = s_{aB}$. Writing explicitly, the Mandelstam invariants are

$$s_{aA} = s_{aB} = -2E k_a^0, \ s_{ab} = 2 k_a \cdot k_b, \ s_{aC} = 4E k_a^0 - \sum_{b=1}^N s_{ab}. \tag{2.6}$$

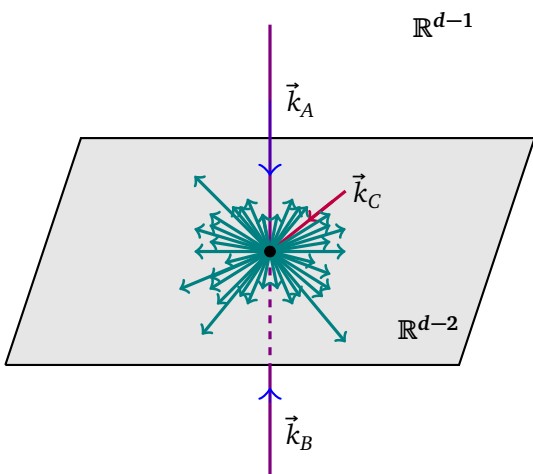

Figure 2.2: The scattering set-up: incoming particles $A$ and $B$ have high energies scaling with $N$ which is the number of the scattered outgoing particles. These $N$ particles all lie on the perpendicular plane to $\vec{k}_A$ and $\vec{k}_B$, with the same energy $k_a^0 = k^0 \; \forall a$ and the same Mandelstam invariants $s_{ab} = s \; \forall \; a, b = 1, \cdots, N$ between them. Particle C will be incoming or outgoing depending on the kinematic variable $\epsilon k^0/s$, in the large N limit.

Three other invariants $s_{AB}$, $s_{BC}$ and $s_{AC}$ won't be important for our discussions in the following. We can determine the angle between two outgoing particles a and b

$$\cos\theta_{ab} = 1 + \frac{s_{ab}}{2|\vec{k}_a||\vec{k}_b|} \, . \tag{2.7}$$

Hence in the physical scattering regime with real $\theta_{ab}$, $s_{ab} \leq 0$. The scattering equations

$$\sum_{j(\neq i)=1}^{N+3} \frac{s_{ij}}{\sigma_i - \sigma_j} = 0 \quad \text{for} \quad i = 1, \cdots, (N+3), \tag{2.8}$$

thus give (ignoring three redundant equations)

$$\frac{2Ek_a^0}{\sigma_a - i} + \frac{2Ek_a^0}{\sigma_a + i} - \frac{s_{aC}}{\sigma_a - \infty} = \sum_{b(\neq a)}^{N} \frac{s_{ab}}{\sigma_a - \sigma_b} \quad \forall a = 1, \cdots, N \, . \tag{2.9}$$

Here we have fixed three punctures $\sigma_A = -i$, $\sigma_B = +i$ and $\sigma_C = \infty$ using the Mobius transformations on the sphere. Now we further specialize to the following kinematic configuration,

1. $s_{ab} = -s < 0, \; \forall \, a, b \in \{1, \cdots, N\}$,

2. $k_a^0 = -k^0 < 0 \; \forall \, a \in \{1, \cdots, N\}$,

3. $E = \frac{1}{4}N\epsilon, \quad \epsilon > 0$ is finite .

With these, $s_{aC} = -N\epsilon k^0 + Ns$ and the particle C is outgoing for $\epsilon > 2k^0$. We also note that the energies of two incoming particles go linearly with the number of particles scattered in the process.

As a consequence of this specialization, the scattering angle between any two of N scattered particles is the same:

$$\cos\theta_{ab} = 1 - \frac{s}{2(k^0)^2} \quad a, b \in \{1, \cdots, N\} , \tag{2.10}$$

and the lengths of the vectors $\vec{k}_a$, $\forall a$ are also the same

$$-|\vec{k}_a| = k_a^0 = k^0 \quad \forall a \in \{1, \cdots, N\}. \tag{2.11}$$

Such a configuration of $N$ vectors $\vec{k}_a$ corresponds to $N$ vertices from the center of a regular

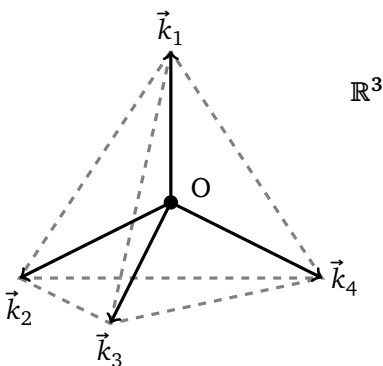

Figure 2.3: Momentum vectors $\vec{k}_1, \cdots, \vec{k}_4$ in our special configuration for $N = 4$ (and hence $d = 5$) form four vertices of a regular tetrahedron in $\mathbb{R}^3$. For higher values of N, they will similarly construct a regular $(N-1)$-simplex in $\mathbb{R}^{N-1}$ with the equal angle $\cos^{-1}(-\frac{1}{N-1})$ in between them.

$(N-1)$-simplex in $\mathbb{R}^{N-1}$. We thus have the restriction on the dimensions of the space-time $\mathbb{R}^{1,d-1}$ as

$$d = N + 1. \tag{2.12}$$

Also the scattering angle between any two vectors from the center to the vertex of such regular simplex is $\cos^{-1}(-\frac{1}{N-1})$, so that from (2.10)

$$\boxed{2\frac{(k^0)^2}{s} = 1 - \frac{1}{N}.} \tag{2.13}$$

In the large $N$ limit, $\frac{(k^0)^2}{s} = \frac{1}{2}$. This condition coming from the simplex structure will play a key role in relating the classical Gross-Mende strings with the graphs generated by the solutions of the scattering equation, see (6.66).

As mentioned in [14], the above set-up can also be considered as a decay process of a very massive particle of mass $M = \frac{1}{2}N\epsilon$ into $(N+1)$ other massless particles indexed with $a$ ($a = 1, \cdots, N$) and $C$. Our analysis in this note is insensitive to such consideration except that the two punctures $A$ and $B$ on the Riemann sphere with a graph will lack any physical interpretation in that case in our later sections.

In terms of the Mandelstam invariants, we are considering the special configuration where

$$s_{aA}, s_{aB} > 0 \quad \text{and} \quad s_{ab} < 0, \tag{2.14}$$

where $\{s_{aA}, s_{aB}, s_{ab}\}$ is the set of independent variables.

In the next few sections, we will take the large $N$ limit to show that the solutions of the scattering equations (2.9) follow a pattern on the sphere $\mathbb{S}^2$ depending on the value of the kinematic parameter $\epsilon k^0/s$.

## 3 Matrix model and Spectral geometry

In this section we will interpret the scattering equation (2.9) as the saddle-point equation of a Penner-like matrix model. Using standard matrix model technology (see [26] for a nice exposition of such techniques), we will then find the corresponding spectral geometry which encodes the solutions of the scattering equations.

**The matrix-model**

The scattering equation (2.9) in our special kinematic configuration becomes,

$$\frac{q}{\sigma_a - i} + \frac{q}{\sigma_a + i} = \frac{2}{N} \sum_{b(\neq a)=1}^{N} \frac{1}{\sigma_a - \sigma_b} \quad \forall a = 1, \cdots, N, \tag{3.15}$$

with

$$q = \epsilon k^0 / s.$$

This notation "q" is inspired from the similarity of (3.15) with the electrostatic problem of three charges located at $-i, +i, \infty$, we will comment further on these soon. In this notation, $s_{aC} = -Ns(q-1)$ and $C$ is outgoing for

$$q > 2 (k^0)^2 / s. \tag{3.16}$$

Interestingly, (3.15) is precisely the saddle-point equation for the "Matrix-model",

$$\mathcal{Z} = \frac{1}{N!} \int \prod_{a=1}^{N} \frac{d\sigma_a}{2\pi} \Delta^2(\sigma_a) e^{-N \sum_{a=1}^{N} W(\sigma_a)}$$
$$= \frac{1}{N!} \int \prod_{a=1}^{N} \frac{d\sigma_a}{2\pi} e^{N^2 \mathcal{S}_{eff}(\{\sigma_a\})}, \tag{3.17}$$

where we have a logarithmic Penner-like potential,

$$W(\sigma) = q \log (\sigma - \sigma_A) + q \log (\sigma - \sigma_B), \tag{3.18}$$

and therefore,

$$W'(\sigma) = \frac{q}{\sigma - \sigma_A} + \frac{q}{\sigma - \sigma_B}. \tag{3.19}$$

$\Delta(\{\sigma_a\})$ in the above expression is the Vandermonde determinant. The effective action is then given by

$$\mathcal{S}_{eff}(\{\sigma_a\}) = -\frac{1}{N} \sum_{a=1}^{N} W(\sigma_a) + \frac{2}{N^2} \sum_{a<b}^{N} \log |\sigma_a - \sigma_b|. \tag{3.20}$$

We can introduce a density of eigenvalues at any $N$,

$$\rho(\sigma) = \frac{1}{N} \sum_{a=1}^{N} \delta(\sigma - \sigma_a), \tag{3.21}$$

which is expected to become a continuous function with support in a compact region of the complex plane in the large N limit. The saddle-point equation then becomes,

$$\frac{1}{2} W'(\sigma) = P \int_{\mathcal{C}} d\sigma' \frac{\rho(\sigma')}{\sigma - \sigma'}. \tag{3.22}$$

We can interpret this problem as a system of $N$ classical charged particles (or $N$ eigenvalues of the matrix model) on the plane moving under the Penner-like external potential (3.18) and logarithmic Coulomb repulsion between them. Such analogies in the context of solutions of the scattering equations have already been mentioned in [14, 15].

For large value of $q$, the external Penner-like potential dominates and the eigenvalues tend to localize at the minimum value $\sigma_*$ of the of the potential. But as $q$ decreases, the Coulomb repulsion will force them to spread over a compact support $\mathcal{C}$. In section 4-5, we will further see a transitions in the patterns of $\mathcal{C}$ on the plane depending on the values of q.

**Spectral Geometry**

Interestingly we can find explicit solutions of the scattering equations written as the saddle-point equations in the corresponding matrix model in the large $N$ limit. Penner-like Matrix models have appeared several times in the past [27–29]. The entire solution to our saddle-point equations is encoded in the spectral curve, defined below. A different method for solving a similar problem using the loop equation has been worked out in [1]. We first define the resolvent via

$$w(z) = \frac{1}{N} \sum_{a=1}^{N} \frac{1}{z - \sigma_a} = \int_{\mathcal{C}} d\sigma \frac{\rho(\sigma)}{z - \sigma_a}. \tag{3.23}$$

Importantly we can read off the density of eigenvalues on $\mathcal{C}$ from this resolvent

$$\rho(\lambda) = -\frac{1}{2\pi i} [w(\lambda + i\epsilon) - w(\lambda - i\epsilon)]. \tag{3.24}$$

We note that $w(z)$ has the following asymptotic behaviour

$$w(z) \sim \frac{1}{z}, \quad z \to \infty. \tag{3.25}$$

The saddle-point equation (3.22) becomes a Riemann-Hilbert problem of determining $w(z)$ from

$$W'(\sigma) = -[w(\sigma + i\epsilon) + w(\sigma - i\epsilon)], \quad \sigma \in \mathcal{C}. \tag{3.26}$$

The solution for this problem [25] yields

$$w_0(z) = \frac{1}{2} \oint_{\mathcal{C}} \frac{dw}{2\pi i} \frac{W'(w)}{z - w} \sqrt{\prod_{k=1}^{2} \frac{z - a_k}{w - a_k}}. \tag{3.27}$$

With our logarithmic potential, we have extra poles at $w = \{z_i\}$ from $W'(w)$ along with at $w = z$ in the integrand of (3.27). We note that the nature of the potential $W'(w) \sim \frac{1}{w}$ removes any pole at infinity coming from the integrand. Thus

$$w_0(z) = \frac{1}{2} \left[ W'(z) - \sum_{i=A}^{B} \frac{q}{(z - \sigma_i)} \sqrt{\prod_{k=1}^{2} \frac{z - a_k}{\sigma_i - a_k}} \right]. \tag{3.28}$$

The full "quantum"-corrected spectral curve is defined in terms of resolvent $w_0(z)$ as,

$$y(z) = W'(z) - 2 w_0(z). \tag{3.29}$$

The spectral curve is thus simply given by,

$$y(z) = \sum_{i=A}^{B} \frac{q}{(z - \sigma_i)} \sqrt{\prod_{k=1}^{2} \frac{z - a_k}{\sigma_i - a_k}}. \tag{3.30}$$

In terms of the spectral curve (3.24) becomes

$$\rho(\lambda) = \frac{1}{4\pi i} \left[ y(\lambda + i\epsilon) - y(\lambda - i\epsilon) \right]. \tag{3.31}$$

i.e, $\rho(\lambda)$ is the discontinuity of $y(z)$ across the branch cut(s) $C$ in between the branch-points $a_1$ and $a_2$.

The branch-points can be determined from imposing the asymptotic behaviour (3.25) of $w_0(z) \sim \frac{1}{z}$ as $z \to \infty$,

$$\sum_{i=A}^{B} \oint_C \frac{dw}{2\pi i} \frac{w^n W'(w)}{\sqrt{\prod_{k=1}^2 (w - a_k)}} = 2\delta_{n,1}. \tag{3.32}$$

This gives two conditions on $\{a_1, a_2\}$:

$$\begin{aligned} f_A + f_B &= 0, \\ f_A \sigma_A + f_B \sigma_B &= 2q - 2. \end{aligned} \tag{3.33}$$

where,

$$f_A = \frac{q}{\sqrt{(\sigma_A - a_1)(\sigma_A - a_2)}}, \quad f_B = \frac{q}{\sqrt{(\sigma_B - a_1)(\sigma_B - a_2)}}. \tag{3.34}$$

We can also rephrase those two constraints in (3.33) as,

$$f_A = \frac{2q - 2}{\sigma_A - \sigma_B}, \quad f_B = \frac{2q - 2}{\sigma_B - \sigma_A}. \tag{3.35}$$

Using these, the spectral curve simplifies (3.30) to,

$$y(z) = (2q - 2) \frac{\sqrt{(z - a_1)(z - a_2)}}{(z^2 + 1)}. \tag{3.36}$$

This is the equation of a complex curve of genus zero. Actually solving (3.33), we can determine $a_1$ and $a_2$

$$a_1 = -a_2 = \sqrt{\frac{q^2}{(1-q)^2} - 1} := c. \tag{3.37}$$

Denoting

$$\alpha = \sqrt{\left| \frac{q^2}{(1-q)^2} - 1 \right|}, \tag{3.38}$$

we note that

$$\boxed{\begin{aligned} c = \alpha \in \mathbb{R}_+ & \quad \text{for } q > \frac{1}{2}, \\ c = i\alpha \in i\mathbb{R}_+ & \quad \text{for } q < \frac{1}{2}. \end{aligned}} \tag{3.39}$$

Thus the branch points rotates by 90° from $(\alpha, -\alpha)$ to $(i\alpha, -i\alpha)$ as we tune $q > \frac{1}{2}$ to $q < \frac{1}{2}$ respectively. This is the key mechanism for transitions in the graphs discussed further in the next few sections.

# 4 Localization on the graphs

In the previous section we have solved the scattering equation in the large N limit in terms the spectral curve (3.36). In this section, this spectral curve will be shown to define the Strebel differential on the sphere with three punctures and so the solutions/eigenvalues of the matrix model localize along the critical Strebel graphs depending on different values of $q$. For a brief introduction to the basic results of Strebel differential, look at appendix A.

The spectral curve (3.36) defines a quadratic differential on the three punctured Riemann sphere (where $\sigma_a$ lives),

$$\phi(z)dz^2 = -\frac{1}{4\pi^2}y^2(z)dz^2 = -\frac{(1-q)^2}{\pi^2}\frac{(z-a_1)(z-a_2)}{(z-i)^2(z+i)^2}dz^2\,, \tag{4.40}$$

which has three double poles at $z = -i, +i$ and $\infty$ and two zeros at $z = a_1, a_2$. The pole at $z = \infty$ can be understood from the change of variables $z = \frac{1}{\eta}$

$$\phi(z)dz^2 = -\frac{(1-q)^2}{\pi^2}\frac{(1-a_1\eta)(1-a_2\eta)}{\eta^2(\eta^2+1)^2}d\eta^2\,, \tag{4.41}$$

which has an explicit double pole at $\eta = 0$.

We can read off the corresponding residues from (3.30) and (4.41) as

$$\oint_A \sqrt{\phi(z)}dz = q, \quad \oint_B \sqrt{\phi(z)}dz = q, \text{ and } \oint_C \sqrt{\phi(z)}dz = 2(1-q)\,, \tag{4.42}$$

which are real and positive for $q < 1$. We will have three distinct cuts of $\phi_S(z)$ $\gamma_1, \gamma_2$ and $\gamma_3$ connecting $a_1$ and $a_2$ on $\mathbb{S}^2$. Using (3.31)

$$\frac{1}{4\pi i}\oint_{\gamma_i} y(z)dz = \int_\gamma dz\rho(z) = l_\gamma\,. \tag{4.43}$$

Again

$$l_\gamma = \int_\gamma \sqrt{\phi(z)}\,dz\,. \tag{4.44}$$

choosing a branch of $\sqrt{\phi(z)}$ where the lengths are positive as indicated by the purple arrows in figure 5.5.

Thus the lengths between $a_1$ and $a_2$ with the metric $\sqrt{\phi(z)}\,dz$ counts the fraction of eigenvalues localized there, hence are always real and positive. The quadratic differential (4.40) thus defines the unique Strebel differential on the sphere with three marked points $A$, $B$ and $C$ with residues at these points as $q$, $q$ and $2(1-q)$ respectively. In fact, putting

$$(L_{(-i)}, L_{(i)}, L_{(\infty)}) = (q, q, 2-2q)\,, \tag{4.45}$$

in the generic form of the Strebel differential with three punctures in (A.89):

$$q = \frac{1}{4\pi^2}\frac{-L_{(\infty)}^2 z^2 - 2i[L_{(i)}^2 - L_{(-i)}^2]z + 2[L_{(i)}^2 + L_{(-i)}^2] - L_{(\infty)}^2}{(z-i)^2(z+i)^2}dz^2\,,$$

we can readily get back (4.40).

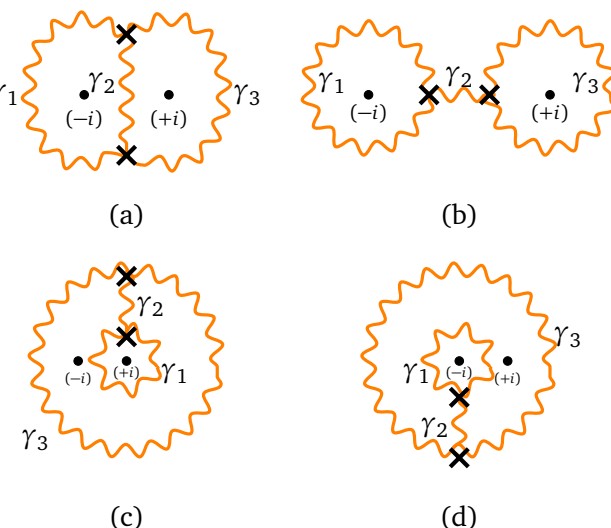

Figure 4.4: Four kinds of Strebel graphs with three punctures depending on the different values of $(L_{(-i)}, L_{(i)}, L_\infty)$.

## Strebel graphs on $\mathbb{CP}^1$ with three punctures

We will take a detour on the Strebel graphs with three punctures before coming back to our main topic of localization of eigenvalues in section 5. Depending on the values of the 3-tuple $(L_{(-i)}, L_{(i)}, L_{(\infty)})$ of the residues of the Strebel differential, there can be four kinds of Strebel graphs with three punctures [31] as shown in Fig. 4.4. We first determine different ranges of residues where the graphs fit in. In all four cases studied below, we have chosen a branch of $\sqrt{\phi(z)}$ such that the lengths of the horizontal trajectories are positive. In the first figure (a),

$$
\begin{aligned}
l_{\gamma_1} + l_{\gamma_2} &= L_{(-i)} && \text{(around } -i\text{)}, \\
l_{\gamma_2} + l_{\gamma_3} &= L_{(i)} && \text{(around } +i\text{)}, \\
l_{\gamma_3} + l_{\gamma_1} &= L_\infty && \text{(around } \infty\text{)}.
\end{aligned}
\tag{4.46}
$$

There is no inequality among $L_{(-i)}$, $L_{(i)}$ and $L_{(\infty)}$ here. But positivity of the lengths imply

$$
\boxed{q < 1.}
\tag{4.47}
$$

In the seconnd figure (b),

$$
l_{\gamma_1} = L_{(-i)}, \quad l_{\gamma_3} = L_{(i)} \quad \text{and} \quad 2l_{\gamma_2} + l_{\gamma_1} + l_{\gamma_3} = L_{(\infty)}.
\tag{4.48}
$$

This implies

$$
L_{(\infty)} > L_{(-i)} + L_{(i)}.
\tag{4.49}
$$

Putting the values of the residues (4.45) for our set-up, this graph corresponds to

$$
\boxed{q < \frac{1}{2}.}
\tag{4.50}
$$

In the third figure (c),

$$
l_{\gamma_1} = L_{(+i)}, \quad l_{\gamma_1} + 2l_{\gamma_2} + l_{\gamma_3} = L_{(-i)} \text{ and } l_{\gamma_3} = L_{(\infty)}.
\tag{4.51}
$$

i.e

$$L_{(-i)} > L_{(i)} + L_{(\infty)}.$$

Similarly in the last figure(d),

$$L_{(i)} > L_{(-i)} + L_{(\infty)}.$$

In our set-up, these two graphs correspond

$$\boxed{q > 1.} \tag{4.52}$$

We end up with two possible graphs for different domains of q: $0 < q < 1/2$, $\frac{1}{2} < q < 1$. We will also comment on the case $q > 1$ in the following.

## 5 Transitions between graphs

We can find a diagnosis for the above mentioned transitions of the Strebel graphs, in our scattering kinematics. From (3.16) and (2.13), C is outgoing for

$$q > 1 - \frac{1}{N}. \tag{5.53}$$

So, in the large N limit, the particle C is always incoming in the domain $0 < q < 1$ (except in the domain $(1 - \frac{1}{N}, 1)$ which is not visible in the strict limit), while it is outgoing for $q > 1$.

$$\boxed{q \gtrless 1 \quad \Leftrightarrow \quad C \text{ outgoing/incoming}.} \tag{5.54}$$

For $q < 1$, the energies of the incoming particles

$$E_A = E_B = E, \quad E_C = (Nk^0 - 2E). \tag{5.55}$$

Hence

$$E_A + E_B - E_C = N(\epsilon - k^0) = N\frac{s}{k^0}\left(q - \frac{1}{2} + \frac{1}{2N}\right) \tag{5.56}$$

In the large $N$ limit, we have the following correspondence in the transition at $q = 1/2$,

$$\boxed{q \gtrless \frac{1}{2} \quad \Leftrightarrow \quad E_A + E_B \gtrless E_C.} \tag{5.57}$$

Next we outline a detailed discussion of these phases in the following.

$\frac{1}{2} < q < 1$ **case:**

Here the zeros of the Strebel differential (3.37) are at $(+\alpha)$ and $(-\alpha)$, with $\alpha \in \mathbb{R}_+$, which clearly fits in with the first graph (a) in Figure 4.4. Solving the equations in (4.46) for the Strebel lengths in the graph type (a),

$$l_{\gamma_1} = l_{\gamma_3} = 1 - q \quad and \quad l_{\gamma_2} = 2q - 1. \tag{5.58}$$

Hence $N$ solutions of the scattering equation localize in three cuts $\gamma_1$, $\gamma_2$ and $\gamma_3$ on the sphere with $(N - Nq)$, $(N - Nq)$ and $(2Nq - N)$ in numbers respectively. Note that for $q < \frac{1}{2}$, $l_{\gamma_2}$ becomes negative and this graph don't naturally extend in such domain.

As clear from the Fig. 5.5, $\sigma_A$, $\sigma_B$ and $\sigma_C$ attract $N$ punctures along the homotopically equivalent Jordon arcs connecting these "charges" $A, B, C$, making those $N$ punctures to localize along the critical Strebel graph of this type.

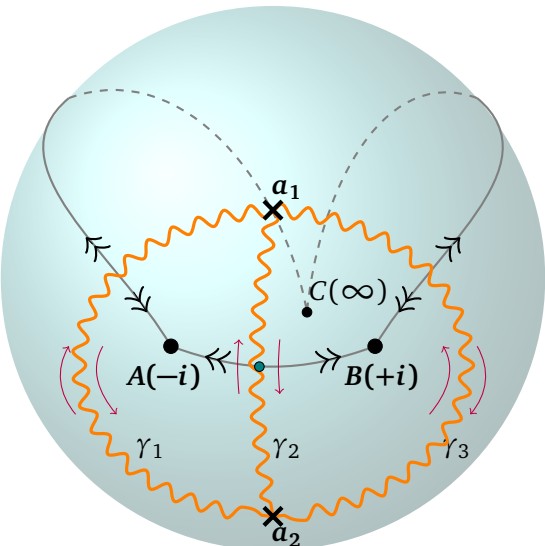

Figure 5.5: In the large $N$ limit, for $\frac{1}{2} < q < 1$ the $N$ punctures localize along the critical Strebel graph which are three cuts of the Strebel differential $\gamma_1$, $\gamma_2$ and $\gamma_3$ between $a_1$ and $a_2$, drawn by orange wavy lines, with the fractions $(1-q)$, $(2q-1)$ and $(1-q)$ respectively. This localization is due to the attractive forces of "charges" $A, B$ and $C$ on those $N$ punctures along the homotopically equivalent Jordan arcs connecting these charges as shown by the double arrows. The choices of the branches and directions for three contour integrations in (4.46) are shown by purple arrows.

### $0 < q < \frac{1}{2}$ case:

The zeros of the differential are now rotated and positioned at $(+i\alpha)$ and $(-i\alpha)$ as required by figure 4.4(b). Solving the equations (4.48) for the second type of graph

$$l_{\gamma_1} = l_{\gamma_3} = q, \quad l_{\gamma_2} = 1 - 2q. \tag{5.59}$$

Thus the solutions localize as before, along the edges of the Strebel graphs $\gamma_1$, $\gamma_2$ and $\gamma_3$ with $Nq$, $Nq$ and $N(1-2q)$ in numbers respectively.

Since $\sigma_C$ has the "charge" $2(1-q)$, the attractive force by $C(\infty)$ is larger compared to the last case for $\frac{1}{2} < q < 1$.

### $q > 1$ case:

In this case, the residue $2(1-q)$ of $y(z)$ at the pole at $z = \infty$ is negative and the corresponding quadratic differential is not a Strebel differential.

It is clear from the matrix model that in this scenario, $\sigma_C = \infty$ repels the $N$ eigenvalues, as its "charge" $2(1-q)$ becomes negative, while $\sigma_A = -i$ and $\sigma_B = i$ still continue to attract them. The resulting configuration takes the shape as in Fig. 5.7, where $N$ punctures localize in the real axis within $[-c, c]$

In particular, we can calculate the density of the punctures $\sigma_a$, $a = 1, \cdots, N$ in the cut $\gamma_2$ using eq. (3.31)

$$\rho(\sigma) = \frac{q-1}{\pi} \frac{\sqrt{c^2 - \sigma^2}}{(\sigma^2 + 1)}, \quad \sigma \in [-c, c]. \tag{5.60}$$

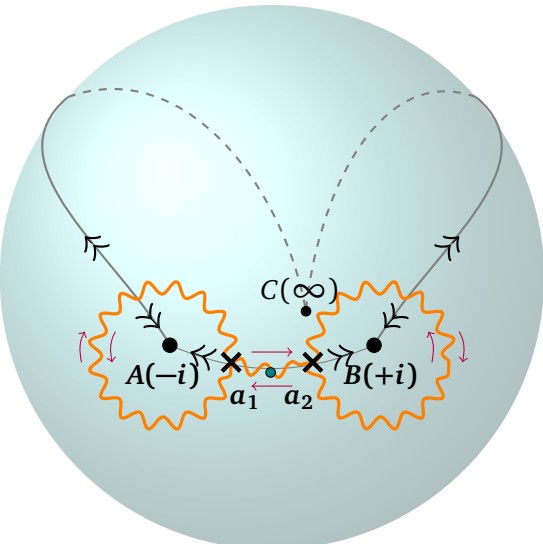

Figure 5.6: In the large $N$ limit, for $0 < q < \frac{1}{2}$ the $N$ punctures localize along the edges $\gamma_1$, $\gamma_2$ and $\gamma_3$ of the Strebel graph, with fractions $q$, $(1-2q)$ and $q$ respectively. As before, three "charges" attract those punctures along the Jordan arcs connecting them.

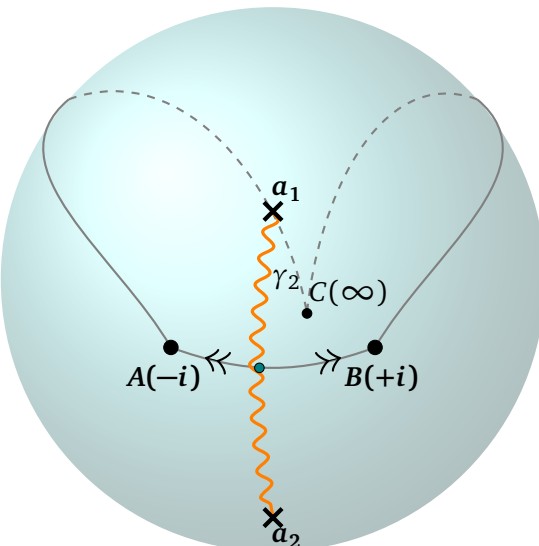

Figure 5.7: For $q > 1$, the $N$ punctures localize, in the large N limit, on the single cut between $a_1$ and $a_2$ shown by the orange wavy line. Here $C$ repels these $N$ punctures, while $A$, $B$ still continue to attract them as in the previous figures.

## 6 Gross-Mende strings and graphs

In this section, we will argue for an explicit relation between the time-component of the classical space-time trajectory of Gross-Mende strings and the quadratic differentials (or the graphs) obtained in the last section for critical space-time dimensions; i.e with $N = 9$ or $25$. As mentioned in appendix B, in the Gross-Mende limit, $p_i^2 \approx 0$ and thus we can directly construct our scattering set-up of section 2 for these strings.

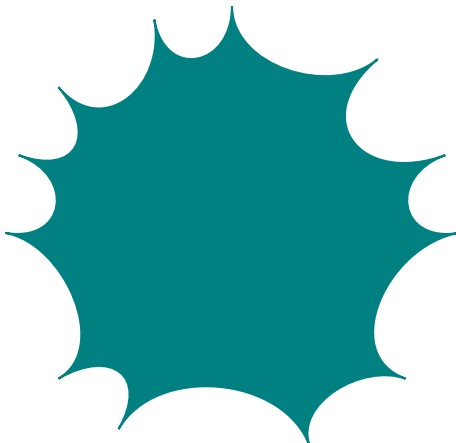

Figure 6.8: Spacetime trajectory of Gross-Mende scattering with 12 strings in d=10.

### Time component of the string trajectory

The classical space-time trajectory corresponding to Gross-Mende scattering with n strings where one worldsheet insertion has been set at $\infty$, is given by [17]

$$X^{\mu}(z) = i \sum_{i=1}^{n-1} k_i^{\mu} G_m(z, z_i), \qquad (6.61)$$

where $G_m(z_1, z_2)$ is defined in appendix B. Note that (6.61) is not reparametrization invariant. At the tree level, the explicit form becomes

$$X^{\mu}(z) = i \sum_{i=1}^{n-1} k_i^{\mu} \log|z - z_i|. \qquad (6.62)$$

In the neighbourhood of the worldsheet insertions, we can use the local co-ordinate $z - z_i \approx e^{-t}$ with $t \to \infty$, so that the classical worldsheet near $z \sim z_i$ becomes

$$X^{\mu}(z) \sim k_i^{\mu} t \quad z \sim z_i. \qquad (6.63)$$

i.e incoming or outgoing strings sweep out a rectilinear motion like a free particle with momentum $k_i^{\mu}$.

It is interesting to ask the interrelation between the graphs and any component of these space-time trajectories. In fact, a priori it is not very clear from the above expression (6.62) that such a connection is explicit, since we have momenta $k_i^{\mu}$ as the coefficients of the logarithms instead of the Mandelstam invariants present in the effective action (3.20) which directly corresponds to the matrix model and the graphs. But we will see, such a connection rather astonishingly comes from the full data of our kinematics. First writing the zeroth component of $X^{\mu}(z)$,

$$-i X^0(z) = -\frac{1}{4} N \epsilon (\log|z| + \log|z-1|) + k^0 \sum_{a=1}^{N} \log|z - z_a|. \qquad (6.64)$$

Now we use the special constraint (2.13) of our kinematics:

$$2\frac{(k^0)^2}{s} = 1 - \frac{1}{N}, \qquad (6.65)$$

which comes from the simplex-structure of the momenta of scattered particles to rewrite $X^0(z)$, in the leading order of $N$, as

$$-i X^0(z) \frac{4k^0}{Ns} \approx -\underbrace{q(\log|z| + \log|z-1|)}_{W(z)} + \frac{2}{N} \sum_{a=1}^{N} \log|z - z_a|$$

$$= -\int^z y(z')dz', \tag{6.66}$$

where we have used the relation

$$\int^z y(z')dz' = W(z) - \frac{2}{N} \sum_{a=1}^{N} \log|z - z_a|, \tag{6.67}$$

up to an additive constant due to unspecified lower limit of the integration. Thus

$$X^0(z) = \frac{\pi Ns}{2k^0} \int^z \sqrt{\phi(z')}\, dz'. \tag{6.68}$$

This explicates the direct connection between $X^0(z)$ of the Gross-Mende strings and the graphs obtained from the Penner-Matrix model.

### Scattering amplitude

We can also relate the scattering amplitude itself with the graphs, since the electrostatic energy in (B.93) for genus zero is simply linked to the effective action of the matrix model (3.20) $S_{eff}$,

$$\alpha' \sum_{i<j}^{n} k_i \cdot k_j \log|z_{ij}| = \frac{\alpha'}{2} Ns \left[ -q \sum_{a=1}^{N} \log|z_a - i| - q \sum_{b=1}^{N} \log|z_b - i| + \frac{2}{N} \sum_{a<b}^{N} \log|z_a - z_b| \right]$$

$$= \frac{\alpha'}{2} sN^2 S_{eff}(\{z_a\}). \tag{6.69}$$

Thus the scattering amplitude computes exponential of the matrix-model potential[1]

$$A_n \sim \exp\left[ \frac{\alpha'}{2} sN^2 S_{eff}(\{z_a^*\}) \right]. \tag{6.70}$$

We can re-express $S_{eff}$ making a connection with the spectral curve

$$\frac{\partial S_{eff}(\{z_c\})}{\partial z_a} = -\frac{1}{N} y(z_a), \tag{6.71}$$

and thus using $y(z) = 2\pi i \sqrt{\phi(z)}$,

$$S_{eff}(\{z_a^*\}) = -\frac{1}{N} \sum_{a=1}^{N} \int^{z_a} y(z)dz$$

$$= -2\pi i \int_{\text{graphs}} d\lambda\, \rho(\lambda) \int^{z_\lambda} \sqrt{\phi(z)}\, dz, \tag{6.72}$$

up to additive constants, where $\{z_\lambda\}$ are the points on the graph. This shows the connection of the amplitudes $A_n$ with the graphs in the leading order of $N$.

---

[1]Here we are not considering the phase factors in (B.99), coming from the Stokes phenomenon in the string amplitudes.

# 7 Discussion

With our very special scattering experiment, we could find the complex solutions of the scattering equations, to localize on the graphs, which then undergo transitions while varying the kinematic parameter $q = \epsilon k^0 / s$. Next we alluded possible connection of these with the Gross-Mende strings for the critical space-time dimensions. Nevertheless, there are a number of problems to be better understood to either enrich or to apply our treatment in this note.

**CHY Scattering Amplitude**

The Cachazo-He-Yuan formula [2–4] for tree-level scattering amplitude of massless particles,

$$\mathcal{A}_n = \int \frac{\prod_{i=1}^n d\sigma_i}{\text{vol}\,SL(2,\mathbb{C})} \prod_i \delta'(k_i \cdot P(\sigma_i)) \mathcal{I}_n(\sigma, k, \epsilon), \tag{7.73}$$

where, $P : \mathbb{CP}^1 \to \mathbb{C}^d$ is a meromorphic map from the Riemann sphere into momentum space,

$$P(\sigma) = \sum_{i=1}^n \frac{k_i}{\sigma - \sigma_i}, \tag{7.74}$$

and so

$$\mathcal{A}_n = \int \frac{\prod_{a=1}^n d\sigma_a}{\text{vol}\,SL(2,\mathbb{C})} \prod_a{}' \delta \left[ \sum_{b(\neq a)} \frac{s_{ab}}{\sigma_a - \sigma_b} \right] \mathcal{I}_n(\sigma, k, \epsilon). \tag{7.75}$$

Thus the moduli integral over the punctured Riemann sphere localizes on the support of the solutions of the scattering equations:

$$\mathcal{A}_n = \sum_{\sigma \in \text{graph}} \frac{\mathcal{I}_n(\sigma, k, \epsilon)}{J(\sigma, k)}. \tag{7.76}$$

It will be interesting to see how the localization of the punctures on the critical Strebel graphs would help to evaluate the above expression (7.76).

**Finite N solution from the roots of the orthogonal polynomials**

In [15], Kalousios considered a special kinematics of scattering of $(N+3)$ particles, where

$$s_{ab} = -1; \quad s_{aA}, s_{aB} < 0,$$

and it was then shown that the $N$ solutions to the scattering equations are precisely the roots of the Jacobi polynomial $P_N^{(\alpha,\beta)}(z)$. We have summarized his main arguments in the following: Jacobi polynomial $P_N^{(\alpha,\beta)}(z)$ obeys the differential equation

$$(1 - x^2)y''(x) + (\beta - \alpha - (\alpha + \beta + 2)x)y'(x) + n(n + \alpha + \beta + 1)y(x) = 0, \tag{7.77}$$

and it has $N$ roots $x_j$ for $j = 1, ..., N$ in the interval $[-1, 1]$

$$P_N^{(\alpha,\beta)}(x) = k \prod_{j=1}^{n-3} (x - x_j). \tag{7.78}$$

It then follows that

$$\sum_{i(\neq j)} \frac{1}{x_i - x_j} = -\frac{(\alpha+1)/2}{(x_i - 1)} - \frac{(\beta+1)/2}{(x_i + 1)}, \quad i = 1, \cdots, N. \tag{7.79}$$

We can then clearly identify $x_a$ with $\sigma_a$ and $s_{aA} = -\frac{\alpha+1}{2}$, $s_{aB} = -\frac{\beta+1}{2}$ for $a = 1, \cdots, N$. One key difference with our set up lies in the fact that the above construction of Jacobi polynomials works only for $\alpha, \beta > -1$ i.e $s_{aA}, s_{aB} < 0$.

It would be very interesting if we can find some judicious orthogonal polynomials applicable in the analogue of the above restricted regime $\alpha, \beta \leq -1$ to explicitly find the $N$ punctures in our set up. We should note that the finite $N$ partition function for Penner-like matrix model (what we have) has been explicitly found in [29].

**Non-critical strings in the Gross-Mende limit**

For non-critical strings, we will have extra contributions from the Liouville sector to the scattering amplitudes with external momenta $\{k_i\}$

$$A_{S_2} \sim \int \prod_{i=1}^{n} d^2 z_i \prod_{a=1}^{m} d^2 w_a \exp\left[-\mathcal{V}(z_i, w_a)\right], \tag{7.80}$$

where the "electrostatic energy" $\mathcal{V}$, similar to (B.93), is given by (with $\alpha'=2$)

$$\mathcal{V}(z_i, w_a) = -\sum_{i,j(i<j)=1}^{n} (k_i \cdot k_j - \beta_i \beta_j) \log|z_i - z_j|^2 + \sum_{j=1}^{n}\sum_{a=1}^{m} \alpha \beta_j \log|z_j - w_a|^2$$
$$+ \alpha^2 \sum_{a,b(a<b)=1}^{m} \log|w_a - w_b|^2. \tag{7.81}$$

For definitions of $\alpha, \beta_j$, see [20]. We mention the structural similarity of the last two terms in (7.81) with the potential of a generic Penner-like matrix model, with suitable values/scaling of $\{\alpha, \beta_j\}$. It is, in fact, in the same spirit of [27], where Penner matrix models appear in the context of Toda theories. It will be interesting to unravel any natural connection with Strebel graphs, for the non-critical strings in the Gross-Mende limit, following our analysis in this note.

# Acknowledgments

I would like to thank Rajesh Gopakumar and Sebastian Mizera for useful discussions and comments on an earlier version of this draft. This work is supported by the Department of Atomic Energy, Government of India, under project no. RTI4001.

# A  Strebel differential

Let $\mathcal{M}_{g,n}$ be the moduli space of the Riemann surfaces with genus $g$ and $n$ punctures. The construction of Strebel differential affords us to write an explicit atlas of $\mathcal{M}_{g,n}$. In this section, we will outline some basic facts about Strebel differential which are relevant for this paper. For more details, see for example [30–33], we will closely follow [32].

**Meromorphic quadratic differential**

A meromorphic quadratic differential $q$ in any complex coordinate chart parameterised by z, on a Riemann surface $\Sigma$ takes the form $\phi(z)dz^2$, where $\phi(z)$ is a meromorphic function of $z$. Under any holomorphic change of coordinates $w = w(z)$,

$$\widetilde{\phi}(w) = \phi(z(w))\left(\frac{dz}{dw}\right)^2. \tag{A.82}$$

We can define a locally flat metric with this quadratic differential

$$ds^2 = |\phi(z)| \, dz d\bar{z} \,, \tag{A.83}$$

which is well defined away from the poles and zeros of $q$.

**Horizontal and vertical trajectories**

We can define two kinds of curves $\gamma(t)$, $t \in (a, b) \subset \mathbb{R}$ on $\Sigma$ classified as

- Horizontal trajectory: $\phi(\gamma(t))\left(\frac{d\gamma(t)}{dt}\right)^2 > 0 \quad \forall \, t \in (a, b) \,,$

- Vertical trajectory: $\phi(\gamma(t))\left(\frac{d\gamma(t)}{dt}\right)^2 < 0 \quad \forall \, t \in (a, b) \,.$

The collection of all horizontal and vertical trajectories foliate $\Sigma$ except at the poles and zeros of the quadratic differential. In the neighbourhood of any regular point $z_0$ on $\Sigma$, we can choose a canonical coordinate

$$w(z) = \int_{z_0}^{z} \sqrt{\phi(z)} \, dz \,, \tag{A.84}$$

so that $q = (dw)^2$. It is then easy to see that the horizontal trajectories near $z_0$ are parallel

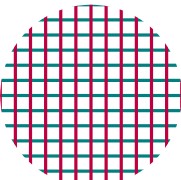

Figure A.9: In the neighbourhood of any regular point of $\Sigma$, the horizontal and vertical trajectories form a rectangular grid like in $\mathbb{C}$.

lines of the real axis

$$\gamma_h(t) = t + ic \,,$$

and the vertical trajectories are those parallel to the imaginary axis

$$\gamma_v(t) = it + c \,,$$

for every $c$ within that neighbourhood.

They behave quite differently though near the poles and zeros:

1. Near a zero of order $m$ at $0$, $q = z^m dz^2$ (up to multiplicative constants), the (m+2) half rays

$$(\gamma_h)_k(t) = t \exp\left(\frac{2\pi i k}{m+2}\right), \quad t \in (0, \infty), \quad \text{for each } k = 0, \cdots, m+1 \,, \tag{A.85}$$

form the horizontal trajectories with one end at $z = 0$. Similarly the $(m+2)$ half rays

$$(\gamma_v)_k(t) = t \exp\left(\frac{2\pi i k + \pi i}{m+2}\right), \quad t \in (0, \infty), \quad \text{for each } k = 0, \cdots, m+1 \,, \tag{A.86}$$

give the vertical trajectories.

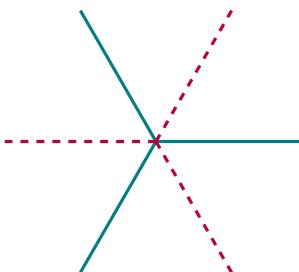

Figure A.10: Horizontal (solid line) and vertical (dashed) trajectories near a zero of order one $q = z(dz)^2$.

2. Near a double pole at 0, $q = -\frac{L^2}{(2\pi)^2}\left(\frac{dz}{z}\right)^2$, cocentric circles with centers at 0 form the horizontal trajectories

$$\gamma_h(t) = r\exp it, \quad t \in \mathbb{R}, \, r > 0, \tag{A.87}$$

while the emergent half-rays from 0

$$\gamma_v(t) = t\exp(i\theta) \quad t > 0, \theta \in [0, 2\pi), \tag{A.88}$$

gives the vertical trajectories.

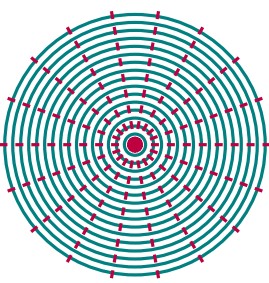

Figure A.11: Horizontal (solid line) and vertical (dashed) trajectories near a double pole: $q = -\frac{L^2}{(2\pi)^2}\left(\frac{dz}{z}\right)^2$.

It is important to note that all the cocentric horizontal trajectories (A.87) have the equal circumferences $L$ with the flat metric (A.83). Also the distance from any such trajectory to the pole is infinite. Near a double pole of the quadratic differential, the geometry of the Riemann surface with the Strebel metric (A.83) takes the form of a semi-infinite cylinder with horizontal trajectories forming level curves (or cross sections of the cylinder) and the vertical trajectories lie parallel to the axis of the cylinder.

A generic horizontal trajectory of any quadratic differential roams around the Riemann surface without closing on itself. But for a special kind of quadratic differentials having only double poles with real and positive residues, all horizontal trajectories are closed except for those which connect zeros of the differential. These compact horizontal leafs foliate the surface into maximal ring domains whose boundaries are formed by non-compact trajectories between the zeros. Such a differential is called a Strebel differential. More concretely, the interesting result [30] of Strebel states,

**Strebel's Theorem**

For every smooth Riemann surface $(\Sigma_g, p_1, \cdots, p_n)$ of genus $g$ with $n$ marked points (punctures) $p_1, \cdots, p_n$ such that $2g + n > 2$ and given an ordered n-tuple $(L_1, \cdots, L_n) \in \mathbb{R}^n_+$, there is a unique quadratic differential $q = \phi_S(z)dz^2$, known as *Strebel differential*, such that

1. $q$ is holomorphic on $\Sigma \backslash \{p_1, \cdots, p_n\}$;

2. $q$ has double pole at any of the marked points $\{p_1, \cdots, p_n\}$;

3. The collection of all non-compact horizontal trajectories is a closed subset of $\Sigma$ of measure zero;

4. Every compact horizontal trajectory is a closed loop $A_i$ centered at $p_i$, such that

$$\oint_{A_i} \sqrt{q} = L_i \,,$$

(choosing the branch of $\sqrt{q}$ so that the integral has a positive value with respect to the positive orientation of $A_i$).

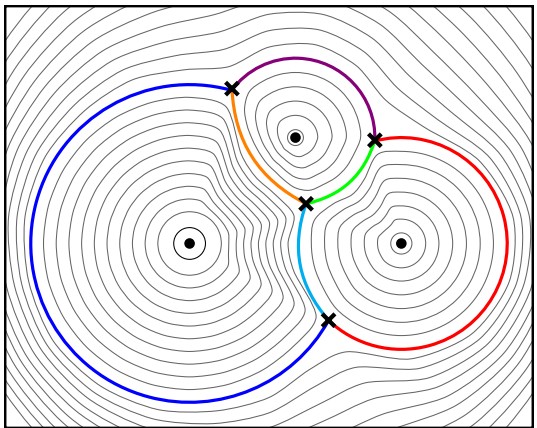

Figure A.12: Horizaontal trajectories of a Strebel differential on a Riemann sphere with four punctures. Black dots and crosses denote the (double) poles and zeros of the Strebel differential. The grey and colored lines describe the compact and non-compact horizontal trajectories respectively. The colored lines form a critical Strebel graph with six edges. This figure is taken from [1].

Let $\Omega_S$ be the set of all Strebel differentials on $(\Sigma_g, p_1, \cdots, p_n)$. We listed some examples in the following:

**Examples**

1. With $(g, n) = (0, 3)$ and $\Sigma = \mathbb{CP}^1$,

$$q = \frac{1}{4\pi^2} \frac{-L_{(\infty)}^2 z^2 - 2i[L_{(i)}^2 - L_{(-i)}^2]z + 2[L_{(i)}^2 + L_{(-i)}^2] - L_{(\infty)}^2}{(z-i)^2(z+i)^2} \, dz^2 \,, \qquad (A.89)$$

where we have fixed three marked points $p_1, p_2$ and $p_3$ at $(-i), +i$ and $\infty$ respectively, using the Mobius transformation on $\mathbb{CP}^1$. We note that this is the unique Strebel differential, or $\dim \Omega_S = 0$.

2. With $(g, n) = (0, n)$ and $\Sigma = \mathbb{CP}^1$,

$$q = -\frac{1}{4\pi^2} \frac{dz^2}{\prod_{j=1}^{n}(z - p_j)} \left( \sum_{i=1}^{n} \frac{L_{p_i}^2 \prod_{j(\neq i)}(p_i - p_j)}{z - p_i} + \sum_{j=0}^{n-4} c_j z^j \right) \,, \qquad (A.90)$$

where $c_i \quad i = 0, \cdots, (n-4)$ are arbitrary complex numbers, i.e $\dim\Omega_S = n-3$.

The above form can be realized remembering the following facts: assuming $q$ doesn't have any pole at $\infty$, $q/(dz)^2$ must be a rational function with $n$ double poles and should behave as $\mathcal{O}(1/z^4)$ as $z \to \infty$ (since there is no pole at $\infty$). Thus $g(z) = \frac{q}{dz^2} \prod_{i=1}^{n}(z-p_i)^2$ must be a polynomial of maximum degree $(2n-4)$ such that $g(p_i) = -L_i^2$. The dimension of the space of such polynomials$=\dim\Omega_S = (2n-3) - n = n-3$.

## B    Gross-Mende Strings

In [17,18], Gross and Mende studied the high energy fixed angle regime of string scattering amplitudes of arbitrary loop, primarily motivated to discover the analogs of short distance physics like operator product expansion, renormalization group in string theory. See [34] for recent progress on the high energy behaviour of scattering amplitudes involving highly excited strings (in contrast to the light strings in Gross-Mende approach).

The g-loop scattering amplitude of tachyons has the following path-integral

$$A_g \sim \prod_i \int d^2 z_i \sqrt{g(z_i)} \int [d\boldsymbol{m}] \exp\left[-\sum k_i \cdot k_j G_{\boldsymbol{m}}(z_i, z_j)\right], \tag{B.91}$$

where $[d\boldsymbol{m}]$ encodes the measure in the moduli space $\boldsymbol{m}$ with all other factors corresponding to the Beltrami differentials and holomorphic differentials for the lacplacian $\nabla^2$ on the worldsheet and $G_{\boldsymbol{m}}(z_i, z_j)$ is the standard Green function for $\nabla^2$:

$$\nabla^2 G_{\boldsymbol{m}}(z_i, z_j) = -\frac{2\pi\alpha'}{\sqrt{g}}\delta^2(z_i, z_j). \tag{B.92}$$

In the high energy, fixed angle regime, the problem reduces to find the extrema of the "electrostatic energy"

$$\mathcal{V}(k_i, z_i, \boldsymbol{m}) = \sum_{i<j} k_i \cdot k_j\, G_{\boldsymbol{m}}(z_i, z_j), \tag{B.93}$$

of 2d Minkowski charges $k_i$ at $z_i$ on a Rieman surface $\Sigma_g$ with moduli parameter $\boldsymbol{m}$, where the surface can change its shape without any energy cost. Since $\alpha' \to \infty$, external states are massless: $k_i^2 \approx 0$, which are important for our analysis in section 6. Also with these massless conditions, $\mathcal{V}$ in (B.93) becomes $SL(2,\mathbb{Z})$ invariant of $\{z_i\}$ [8].

Though we only discussed bosonic string throughout this section, the high energy behaviour of string amplitudes are identical for superstrings as well. In particular the exponential behaviour in (B.91) is common to any string theory.

**Toy example**

We can understand the basic treatment in a toy model of tree level amplitudes of four tachyons in the closed string theory. Explicitly it has the standard expression [23]

$$A_{S_2}(k_1, k_2, k_3, k_4) \sim \frac{\Gamma(-1 + \frac{\alpha's}{4})\Gamma(-1 + \frac{\alpha't}{4})\Gamma(-1 + \frac{\alpha'u}{4})}{\Gamma(2 - \frac{\alpha's}{4})\Gamma(2 - \frac{\alpha't}{4})\Gamma(2 - \frac{\alpha'u}{4})}, \tag{B.94}$$

where $s = (k_1 + k_2)^2$, $t = (k_1 + k_3)^2$ $u = (k_1 + k_4)^2$ and in the center of mass frame, $s = -E^2$, $t = (E^2 + 16/\alpha')\sin^2(\theta/2)$, $u = (E^2 + 16/\alpha')\cos^2(\theta/2)$ with E being the center-of-mass energy and $\theta$ is the angle between the particle 1 and 3. Note that $s + t + u = +16/\alpha'$.

In the large $E$ and fixed $\theta$ limit (or equivalently large $s$ and fixed $t/s$), we can use Stirling approximation for Gamma functions to readily get

$$A_{S_2} \sim \exp\left[\frac{\alpha'}{2}(s\log(s\alpha') + t\log(t\alpha') + u\log(u\alpha'))\right]. \tag{B.95}$$

We can argue the above form by a saddle-point calculation as well, because in a different approach, the scattering amplitude comes from the worldsheet integral

$$A_{S_2} \sim \int_{\mathbb{C}} d^2z_4 \, |z_4|^{\alpha'u/2-4} |1-z_4|^{\alpha't/2-4}, \tag{B.96}$$

where three other insertion points on the worldsheet are fixed by the Mobius transformation on the sphere: $z_1 = 0, z_2 = 1, z_4 \to \infty$. In the large $\alpha'$ limit (which is the same as high-energy fixed angle limit, since $\sin^2(\theta/2) = -\frac{\alpha't}{\alpha's-16}$) we can perform a saddle-point analysis of the exponential

$$-\frac{\alpha'u}{2}\log|z_4| - \frac{\alpha't}{2}\log|1-z_4|, \tag{B.97}$$

with respect to $z_4$, to obtain the critical values

$$|z_4^*| = -\frac{u}{s} \quad |1-z_4^*| = -\frac{t}{s}, \tag{B.98}$$

so that the saddle-point value $A_{S_2}(z_4^*)$ reduces to the same expression as (B.95).

We should mention that this simple-looking saddle-point analysis is, in fact, not correct; in particular, there is a Stokes phenomenon[2] [21] involved in (B.94), where the asymptotic limit of (B.94) depends on the direction of approaching $\alpha' \to \infty$ limit. Clearly we will have an infinite number of poles if any of $\mathcal{R}(s), \mathcal{R}(t), \mathcal{R}(u)$ becomes less than $4/\alpha'$ in approaching the high-energy fixed angle limit. The way to realise this from (B.96) is to note that the integrand is really defined on an infinite-sheeted surface $\widetilde{\mathcal{M}}_{0,4}$, which is the universal cover of the moduli space $\mathcal{M}_{0,4} = \{z \in \mathbb{CP}^1 | z \neq 0, 1, \infty\}$ with saddle points from each sheet. Interestingly these infinite number of saddle-contributions can be resummed only to yield an oscillatroy factor [19, 22]

$$A_{S_2} \sim \frac{\sin(\pi\alpha't)\sin(\pi\alpha'u)}{\sin(\pi\alpha's)} \exp\left[\frac{\alpha'}{2}(s\log(s\alpha') + t\log(t\alpha') + u\log(u\alpha'))\right]. \tag{B.99}$$

In this note, we will mainly be interested in the second exponential part.

**For genus zero**

The worldsheet correlators for genenric vertex operator insertions are given by the following form, which is further required to integrate over the moduli of the worldsheet (for sphere these are simply the insertion points) [23],

$$\left\langle \prod_{i=1}^{n} \left[e^{ik_i \cdot X(z_i, \bar{z}_i)}\right]_r \prod_{j=1}^{p} \partial X^{\mu}(z_j') \prod_{k=1}^{q} \bar{\partial} X^{\nu_k}(\bar{z}_k'') \right\rangle_{S_2}$$

$$= iC_{S_2}^X (2\pi)^d \delta^d\left(\sum_i k_i\right) \exp\left[\alpha' \sum_{i<j}^{n} k_i \cdot k_j \log|z_{ij}|\right] \tag{B.100}$$

$$\times \left\langle \prod_{j=1}^{p} [y^{\mu_j}(z_j') + q^{\mu_j}(z_j')] \prod_{k=1}^{q} [\tilde{y}^{\nu_k}(\bar{z}_k'') + \tilde{q}^{\nu_k}(\bar{z}_k'')] \right\rangle,$$

---

[2]I thank Sebastian Mizera for pointing this out to me.

where

$$y^{\mu}(z) = \langle \partial X^{\mu}(z) \prod_{i=1}^{n} e^{ik_i \cdot X(z_i)} \rangle = -i\frac{\alpha'}{2} \sum_{i=1}^{n} \frac{k_i^{\mu}}{z - z_i},$$

$$\tilde{y}^{\mu}(\bar{z}) = \langle \bar{\partial} X^{\mu}(\bar{z}) \prod_{i=1}^{n} e^{ik_i \cdot X(\bar{z}_i)} \rangle = -i\frac{\alpha'}{2} \sum_{j=1}^{n} \frac{k_i^{\mu}}{\bar{z} - \bar{z}_i}, \tag{B.101}$$

and $q^{\mu} = \partial X^{\mu} - y^{\mu}$.

In $\alpha' \to \infty$ limit, we get the saddle-point equations

$$\sum_{j(\neq 1)}^{n} \frac{k_i \cdot k_j}{z_i - z_j} = 0 \quad \text{for} \quad i = 1, \cdots, n, \tag{B.102}$$

which are known as scattering equations. We also note that the Green function for $g = 0$ is

$$G(z_i, z_j) = -\frac{\alpha'}{2} \log |z_{ij}|^2. \tag{B.103}$$

**For higher genus**

We will closely follow [17] in the following discussion. The scattering amplitude at $g$-loop can be approximated by a sequence of $(g+1)$ elastic scatterings with the same center-of-mass energies $s$ and momentum transfer $-t_i, i = 1, \cdots, g+1$

$$A_g \approx \text{Max}\left[ s^{-g} \prod_{i=1}^{g+1} A_{tree}(s, t_i) \right], \tag{B.104}$$

where we have the factor $s^{-g}$ from $g$-body phase space with further constraints $\sum_{i=1}^{g-1} \sqrt{-t_i} \leq \sqrt{-t}$. Unlike power-law fall-off for field theory, the string amplitudes damps off exponentially: $\exp[-s f(\phi)]$, $\phi$ being the scattering angle for the 4 particle process. In the small scattering regime, if $f(\phi)$ behaves as $\phi^p$, we confront with the following extremization problem

$$s^{-g} \exp[-s \sum_{i=1}^{g+1} \phi_i^p] \quad \text{with} \quad \sum_{i=1}^{g+1} \phi_i \leq \phi = 2\sqrt{-t/s}. \tag{B.105}$$

The maximum is achieved when all $\phi_i$ s are equal to $\phi/(g+1)$, so that, in the small $\phi$ approximation

$$A_g \approx A_{\text{tree}}^{1/(g+1)}. \tag{B.106}$$

This also implies that high energy behaviour of string scattering is dominated by hard scattering, i.e the intermediate momentum transfers are large, so that the scattering integrals have a saddle-point, which we can, in fact, determine at each order in the perturbation theory.

Let's digress a bit into the higher genus surfaces. A higher genus surface with $Z_M$ automorphism has the form

$$y^M = \prod_{i=1}^{n} (z - z_i)^{n_i}, \tag{B.107}$$

where $n_i$ are relatively prime to $M$, so that it represents a $M$-sheeted Riemann surface with $n$ branch points at $a_i$ each with order $(M-1)$. From Riemann-Hurwitz formula, the genus of the surface is simply

$$g = 1 - M + \frac{1}{2} \sum_{i=1}^{n} (M-1) = \frac{1}{2}(M-1)(n-2). \tag{B.108}$$

Now we place the charges $k_i$, $i = 1, \cdots, n$ on the branch points which are separated by $1/M$ times a period. Since any of the $M$ sheets perceives a branch point in the same way, the electric field produced by these charges

$$\mathcal{E}^{\mu} = \frac{1}{M}\mathcal{E}_1^{\mu} = \frac{1}{M}\sum_{i=1}^{n}\frac{p_i^{\mu}}{z - z_i}, \tag{B.109}$$

where $\mathcal{E}_1^{\mu}$ is the electric field if all those charges would be placed on a single sheet. The electrostatic energy in this configuration becomes

$$\mathcal{V}_M = \frac{1}{M}\mathcal{V}_1 = -\frac{1}{2M}\sum_{i<j}k_i \cdot k_j \log|z_i - z_j|. \tag{B.110}$$

Thus these $Z_M$ curves have the correct exponential behaviour of being $M$-th root of the tree diagram, similar to (B.106), and they have higher symmetry like $Z_M$, which is somewhat expected for the high energy strings. With such arguments, [17,18] alluded them to be the dominant saddle-point contribution for genus $g = \frac{1}{2}(M-1)(n-2)$ diagram of the string scattering.

In particular, an interesting lesson from this discussion is the applicability of the scattering equations (2.9) for higher genus string amplitudes as well, as obtained from extremizing (B.110).

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
