# Peer review of "Scattering and Strebel graphs"

_SciPost Physics, doi:SciPost Phys. 13, 010 (2022)_

## Round 1 · Referee Report · Anonymous (Referee 1) · 2022-5-22

Strengths

The idea and results presented in the paper ate novel and original.

Weaknesses

The discussion in this paper is limited to a very special type of kinematics. It is unclear whether the results of this paper can improve our understanding of scattering amplitudes in field and string theories.

Report

Scattering equations, a system of algebraic equations relating scattering data with the moduli space of punctured Riemann spheres, provide a novel way to reformulate the tree-level S-matrix in many massless quantum field theories. This paper discusses a special type of kinematics, i.e. n particles in n-2 spacetime dimensions. Using these data, the author found an interesting relation between the (solutions of) scattering equations and a matrix model and the so-called Strebel graphs.

The results are new and very well-presented. It should be accepted in SciPost Physics.

Requested changes

  1. On page 3, it should have a clear statement how the on-shell conditions are satisfied.

  2. A typo in eq(2.5), $\ne 1$ should be $\ne i$. In this formula, it should be more clear for $n (=N+3)$.

  3. On page 4, regarding the scattering equation system, it should be clear that one deleted three redundant equations and fix three punctures $\sigma_{A,B,C}$ explicitly.

  • validity: high
  • significance: ok
  • originality: high
  • clarity: high
  • formatting: excellent
  • grammar: excellent

Author:  Pronobesh Maity  on 2022-06-18  [id 2589]

(in reply to Report 1 on 2022-05-22)
Category:
remark
correction

I would like to thank the referee for his/her useful comments and for recommending publication of the paper. I have implemented the suggested changes in the resubmitted file.

---

## Round 1 · Referee Report · Anonymous (Referee 3) · 2022-5-23

Strengths

The results in this paper are interesting and novel.

Weaknesses

The results of this paper apply only for very special kinematics.

Report

In this article, the author has obtained an interesting connection between solutions of scattering equations and Strebel graphs. In particular, the author considers the scattering of $n$ massless particles in a special kinematic regime where two of the incoming particles have energies that scale with $n$. In this setting, the author shows that the solutions of the scattering equations can be mapped to the saddle point equations of a matrix model in the limit where the number of particles is large. In terms of the matrix model, it is then shown that the solutions to the scattering equations localize on Strebel graphs which are determined in terms of the spectral curve for the above matrix model. The nature of the localization also changes in an interesting fashion as a kinematic parameter, labeled by $q$ in the paper, is varied. The author also points out a connection between Strebel graphs and the high energy, fixed angle limit of string scattering amplitudes.

In my opinion, the central results presented here are interesting and I recommend the paper for publication.

Requested changes

I suggest the author fix the following typo:

  1. In the second line of equation (3.3), the Vandermonde determinant should no longer be there in the measure, since it is meant to be included in the definition of the effective action.

  • validity: good
  • significance: good
  • originality: high
  • clarity: good
  • formatting: good
  • grammar: good

Author:  Pronobesh Maity  on 2022-06-18  [id 2591]

(in reply to Report 3 on 2022-05-23)

I would like to thank the referee for his/her useful comments and for recommending publication of the paper. I have implemented the suggested changes in the resubmitted file.

---

## Round 1 · Referee Report · Anonymous (Referee 2) · 2022-5-23

Strengths

Uncovers a connection between solutions of tree-level scattering equations for special kinematics and Strebel graphs.

Weaknesses

The kinematic configurations studied in this paper cannot be realized in a fixed number of space-time dimensions when the number of external particles goes to infinity.

Report

This submission discusses the connection between worldsheet approaches to tree-level scattering amplitudes and Strebel graphs. The central role in this work is played by "scattering equations", which are a system of rational equations determining positions of punctures (positions of vertex operators) on a Riemann sphere for a given choice of the kinematics of external particles (but independent of other quantum numbers, such as polarizations and colors). They can be thought of as determining the geometry of string worldsheet in the classical limit, as saddle point equations in the $\alpha' \to \infty$ limit, but also feature in the Cachazo-He-Yuan formalism for computing field-theory amplitudes that localize on the same points in the moduli space of punctured Riemann spheres. The essential question studied in this work is what happens to these localization points in the limit as the number of external states becomes infinite.

To this end, the author chooses a specific kinematic configuration for massless scattering in $d$ space-time dimensions, which looks like a high-energy forward scattering of two particles into $N-2$ soft ones. It is symmetric among the $N-3$ out of the $N-2$ particles, which vastly simplifies the analysis of scattering equations. In order to realize this kinematics for large $N$, the number of dimensions $d$ needs to be large too. On top of $N$, this kinematics is parametrized by a positive variable $q$. The author considers the large-$N$ limit of scattering equations for different values of $q$.

The central result of this work is that in the large-$N$ limit, one can map the scattering question to a resolvent problem, not dissimilar to the ones encountered in Penner matrix models. This allows the author to show that solution of scattering equations arrange themselves along certain graphs, the Strebel graphs, on the Riemann sphere. Surprisingly, the specific topology of this graph depends discontinuously on the value of $q$: there are "phase transitions" at $q = 1/2$ and $q = 1$, which the author studies in the text. The first one is expected by the fact that one of the energies flips the sign, but the second one appears to remain unexplained. It is an interesting result overall.

As for the applications of this result, the most promising seems to be studying the geometry of the string worldsheet in the high-energy, large-$N$ limit, which is mentioned in Sec. 6. In principle, the are two obstacles with immediately applying the above analysis to this problem: that the kinematics cannot be realized in a fixed number of space-time dimensions; and that saddles the author finds are complex. The first one can be only cured with a more complicated (less symmetric) choice of kinematic configurations, but at this stage it is not clear one can leverage the connection to matrix models, which depended on the symmetry in the first place. The second question concerns the fact that solutions of scattering equations become complex, which means one needs to carefully study how the integration contour passes through such saddles to determine which of them are relevant, in addition to computing the phase weight of each saddle. These remain interesting problems for the future.

The material is already presented well and is suitable for publication, but below I give a few minor comments that should be addressed first.

In conclusion, this article contains new and interesting results in the area of worldsheet methods for scattering amplitudes. Therefore, I recommend it for publication in SciPost Physics.

Requested changes

  • In the introduction, the author gives a list of papers on solving scattering equations numerically, which should include the currently most efficient algorithm, described in the reference https://inspirehep.net/literature/1835427.

  • Above eq. (2.1), one might want to use $\mathbb{R}^{1,d-1}$ instead of $\mathbb{R}^{d-1,1}$ to avoid the ambiguity of which momentum component is the energy. Similar corrections should be made in the rest of the paper for consistency.

  • In the bottom limit in eq. (2.6) $j(!=1)$ should've been $j(!=i)$.

  • In eq. (3.13) and (3.18), the integration contour should be $\mathcal{C}$ instead of $C$.

  • Eq. (6.1) and (6.2) are exact classical solutions, so there should be no corrections included on the RHS. Even if they were included, they should be subleading in $\alpha'$ (not $s$). The phrase "one worldsheet insertion has been set at $\infty$" also doesn't make sense beyond genus zero, so I suggest removing it and make the summation go to $n$ instead of $n-1$.

  • In eq. (B.3), the range of the sum should be $i<j$ instead of $i \leq j$.

  • validity: high
  • significance: good
  • originality: high
  • clarity: good
  • formatting: reasonable
  • grammar: good

Author:  Pronobesh Maity  on 2022-06-18  [id 2590]

(in reply to Report 2 on 2022-05-23)

I would like to thank the referee for his/her useful comments and for recommending publication of the paper. I have implemented the suggested changes in the resubmitted file.

---

## Round 2 · List of Changes

The following changes have been made:

  1. Reference [13] https://inspirehep.net/literature/1835427 has been added.

  2. $\mathbb{R}^{d-1,1}$ has been replaced by $\mathbb{R}^{1,d-1}$ throughout the text.

  3. In eq. (2.6), $j(\neq 1)=1$ has been replaced by $j(\neq i)=1$.

  4. In eq. (3.13) and (3.18), integration contour C has been replaced by $\mathcal{C}.

  5. In eq. (6.1) and (6.2), $\mathcal{O}(1/s)$ has been removed.

  6. In eq (B.3), $i \leq j$ is replaced by $i<j$.

  7. In the 2nd line of eq. (3.3), the Vandermonde determinant $\Delta^2({\sigma_a})$ has been removed.

  8. Above eq. (2.7) we mentioned that three redundant scattering eq.s are removed and below the eq. we specified the choices of $\sigma_{A,B,C}$ explicitly.

  9. Below eq. (2.3), we mentioned that the on-shell condition is satisfied.

  10. In eq. (2.9), we added a -ve sign on the L.H.S.

  11. Added a line "But a good analytical method is still missing" in the 2nd stanza of page 1.

  12. Corrected the typo from "chqnge" to "change" below eq. (B.3).

---

## Editorial Decision

published